# HFR-Video-Based Stereo Correspondence Using High Synchronous Short-Term Velocities

**DOI:** 10.3390/s23094285

**Published:** 2023-04-26

**Authors:** Qing Li, Shaopeng Hu, Kohei Shimasaki, Idaku Ishii

**Affiliations:** Smart Robotics Laboratory, Graduate School of Advanced Science and Engineering, Hiroshima University, 1-4-1 Kagamiyama, Higashi-Hiroshima, Hiroshima 739-8527, Japan

**Keywords:** high-speed vision, stereo correspondence, motion information, high synchronous velocity

## Abstract

This study focuses on solving the correspondence problem of multiple moving objects with similar appearances in stereoscopic videos. Specifically, we address the multi-camera correspondence problem by taking into account the pixel-level and feature-level stereo correspondences, and object-level cross-camera multiple object correspondence. Most correspondence algorithms rely on texture and color information of the stereo images, making it challenging to distinguish between similar-looking objects, such as ballet dancers and corporate employees wearing similar dresses, or farm animals such as chickens, ducks, and cows. However, by leveraging the low latency and high synchronization of high-speed cameras, we can perceive the phase and frequency differences between the movements of similar-looking objects. In this study, we propose using short-term velocities (STVs) of objects as motion features to determine the correspondence of multiple objects by calculating the similarity of STVs. To validate our approach, we conducted stereo correspondence experiments using markers attached to a metronome and natural hand movements to simulate simple and complex motion scenes. The experimental results demonstrate that our method achieved good performance in stereo correspondence.

## 1. Introduction

Stereo vision offers a straightforward way for computers to comprehend the world and can reconstruct the three-dimensional geometric information of scenes [1]. It is widely used in various fields such as autonomous navigation systems for mobile robots [2], aerial and remote sensing measurements [3], medical imaging [4], SLAM [5], and more. Stereo correspondence is a crucial element of stereo vision that plays a vital role in finding corresponding point pairs between two images to calculate the depth information of the stereo image [6].

The goal of this study is to achieve stereo correspondence for multiple moving objects with similar appearances. Over the past few decades, extensive research has been dedicated to stereo correspondence. Traditional stereo correspondence algorithms can be categorized into local, global, and semi-global methods. These methods use manually extracted features, such as sum of absolute difference (SAD) [7], normalized cross-correlation (NCC) [8], SIFT (Scale-Invariant Feature Transform) [9], and ORB (Oriented FAST and Rotated Brief) [10], to provide similarity measures between left and right image patches. However, the performance of traditional stereo correspondence methods is severely limited by the handcrafted features used in the cost function. In Ref. [11], convolutional neural networks (CNNs) were first introduced for stereo correspondence, demonstrating advantages in both speed and accuracy over traditional methods. Currently, deep learning-based image similarity measurement methods mainly rely on feature extraction from deep networks [12] and similarity comparison through metric learning [13].

However, appearance-based correspondence methods face significant challenges due to variations in camera viewpoints, lighting conditions, and pose changes [14]. Motion information, on the other hand, is independent of object appearance and exhibits excellent performance in scenes with similar appearances and drastic changes in appearance. Currently, a significant amount of research has been devoted to cross-camera multi-object correspondence based on motion information [15,16]. Existing motion similarity measurement methods can be divided into two categories: spatial similarity and spatio-temporal similarity [17]. Spatial similarity only considers the same geometric shape and ignores the temporal dimension, which is not suitable for real-time stereo correspondence systems. The update of motion information is delayed due to the limited speed of traditional visual image input (30 or 60 fps) [18], making trajectory synchronization of high-speed moving objects difficult. However, high-speed vision sensors operate at hundreds or even higher frequencies, enabling them to observe moving objects and capture phase differences with extremely low latency [19]. Additionally, viewing angles significantly affect trajectory matching performance. First-order motion velocity and second-order acceleration directions are relatively insensitive to viewing angles.

In this study, we propose a high-speed stereo correspondence system for multiple moving objects with similar appearances, based on their high synchronous velocities. We designed stereo correspondence experiments for moving objects with different frequencies and amplitudes, as well as for high-speed moving hands with occlusions. The subsequent parts of this study are organized as follows: related works and research are presented in Section 2; a detailed algorithm analysis and concept illustration are presented in Section 3; Section 4 presents a full description of the experimental test platform, followed by a discussion of the test results; and finally, the conclusions are presented in Section 5.

## 2. Related Works

This study aims to correspond multiple moving objects with similar appearances in a stereoscopic video, which is closely related to research on image similarity measurement and trajectory similarity measurement. In the following sections, we will provide a brief review of related works.

### 2.1. Image Similarity Measurement

The computation of image matching serves as the initial step in stereo correspondence, relying primarily on the similarity of target pixel blocks surrounding the stereo images. Over time, the measurement of similarity between image blocks has evolved from region-based approaches to feature-based approaches, and finally to deep learning techniques.

Region-based matching methods can be classified into two categories. The first approach minimizes differences in pixel information by using methods such as cross-correlation [20], mean square error (MSE) [21], and mutual information [22]. The second approach transforms images from the time domain to the frequency domain and performs similarity analysis in the transformed domain using techniques such as Fourier transform [23], Walsh transform [24], and wavelet transform [25]. However, region-based image matching methods require high-quality images because noise, lighting, and changes in shape can greatly affect the quality of the match. Feature-based methods can significantly reduce the impact of image quality on similarity and have been extensively researched to date [26]. These features are often manually designed, such as SURF [27], ORB [28], and LBP [29]. Feature-based methods require additional computational power to find matching points with similar features between image blocks. The Structural Similarity Index (SSIM) combines brightness, contrast, and structure to achieve matching results similar to human visual perception and has been widely used for comparing image similarity [30].

Recently, convolutional neural networks (CNNs) have replicated the huge success in image recognition and have become a research hotspot in image region matching. Based on CNNs, image matching can be mainly divided into two research directions: (1) using deep networks such as ResNet [31] and VGG [32] to extract image features and then using similarity metrics such as Euclidean distance and cosine distance to measure the similarity of high-dimensional features; (2) using metric learning to directly output the similarity of two image blocks. In Ref. [33], the ResNet model was used to extract periocular features from different spectral bands, and cosine similarity was used for image verification, achieving high accuracy. In Ref. [34], the VGGNet was used to extract multi-scale features from segmented patches and achieved detection of forged images. Compared to manually extracted features, features extracted by CNNs are more effective in handling noise and morphological changes. In Ref. [35], MatchNet was proposed, which uses CNN for region feature extraction and then computes similarity through a three-layer fully connected network. The DeepCompare method was proposed in Ref. [36], which improved the performance of the Siamese network using the Center-Surround Two-Stream Network and Spatial Pyramid Pooling (SPP) [37]. DeepCD based on the Triplet network was proposed in Ref. [38]. This method describes image patches as complementary descriptors and improves the performance in various applications. Currently, methods based on deep learning are difficult to output calculation results in extreme time and are not suitable for high-speed vision systems. However, the matching performance they provide is unmatched by traditional algorithms.

### 2.2. Matching Based on Motion

When objects are well tracked under good conditions of a single camera, their motion information is less affected by lighting, shape changes, and noise. Motion-based matching has been widely used in cross-camera multi-object matching, such as in smart traffic [39], user behavior analysis [40], and motion pose estimation [41].

There are various ways to represent motion information, such as trajectories, angles, and velocities. Trajectories, as an easily obtainable form of motion information, have been widely used in multi-object tracking. Trajectories can be classified into two types: sequence-only trajectories and spatiotemporal trajectories, depending on whether the temporal property is considered [42].

Different methods have been developed for measuring the similarity between different target trajectories, which are mainly divided into three directions: distance-based, feature-based, and deep learning-based trajectory similarity calculation methods. Distance-based trajectory similarity calculation methods mainly measure the similarity between trajectories by calculating the distance between trajectory points. Some classic methods include Dynamic Time Warping (DTW) [43], Edit Distance on Real sequence (EDR) [44], and Longest Common Subsequence (LCSS) [45]. For instance, LCSS is used to calculate the similarity of the 3D GPS trajectories of the trucks in Ref. [46] to identify the movement patterns of the trucks. In Ref. [47], a trajectory evaluation method based on Dynamic Time Warping was proposed to evaluate the discrepancy between robot trajectories and human motion. However, these methods have limitations in dealing with data noise and missing values.

Feature-based trajectory similarity calculation methods extract features from trajectories and then calculate the similarity between features to measure the similarity between trajectories. Some classic methods include Shape Context [48], Histogram of Oriented Gradients (HOG) [49], and Global Alignment Kernel (GAK) [50]. For example, a skeleton-based action recognition method is proposed in Ref. [51], which combines trajectory images and visual features to simulate human actions. Based on the Fréchet distance, a shape-based local spatial association metric is proposed in Ref. [52] for detecting anomalous activities of moving ships. However, these methods are more complex in feature extraction and computation, and require a larger amount of computation.

Deep learning-based trajectory similarity calculation methods use machine learning to model and learn trajectory data, and then calculate the similarity between trajectories. Some classic methods include neural network-based methods, decision tree-based methods [53], and support vector machine-based methods [54]. For instance, an RNN-based Seq2Seq autoencoder model is proposed in Ref. [55], which improves the calculation of similarity. In Ref. [56], an attention-based robust autoencoder model is proposed, which learns low-dimensional representations of noisy ship trajectories. An unsupervised learning method is proposed in Ref. [57], which can automatically extract low-dimensional data features through convolutional autoencoders. The similarity between trajectories can be obtained from the similarity between low-dimensional data, which ensures high-quality trajectory clustering performance. However, these methods require a large amount of training data and computation resources, but they offer higher accuracy and robustness in trajectory similarity calculation.

## 3. HFR Stereo Correspondence Based on High Synchronous Short-Term Velocities

### 3.1. Concept

As mentioned in previous sections, matching multiple moving objects with similar appearances in stereoscopic video is a significant challenge. To address this issue, we propose a High Frame Rate (HFR) stereo vision system, as depicted in Figure 1. The entire process of stereo correspondence for multiple objects is divided into two steps: independent multiple-object tracking and stereo correspondence based on high synchronous short-term velocities. In the independent multiple-object tracking step, we define the pixel-scale movement of an object between HFR frames as its velocity, which comprises horizontal and vertical components. As shown in Figure 1a, we utilize *n* velocities over a period of time before the current time as the motion feature of the objects, referred to as short-term velocities. In the object stereo correspondence step, we analyze the similarity between the high synchronous short-term velocities of multiple objects frame-by-frame to establish correspondences among different objects, as illustrated in Figure 1b.

### 3.2. Independent Multiple-Object Tracking in HFR Stereoscopic Video

In this study, we conducted offline experiments using the HFR stereoscopic videos to validate the effectiveness of our algorithm. The first step involves the fast tracking of multiple objects using the HFR stereo camera, which enables the real-time update of the motion positions and velocities of the objects. However, HFR stereoscopic videos not only provide more image information but also impose a higher computational burden on multiple object tracking. HFR stereoscopic videos usually run at 200 frames per second or higher, leaving us with only 5 milliseconds or less for computation. However, detectors that yield good detection performance usually require longer running times. For instance, in this study, the hand detection using MediaPipe takes approximately 30 milliseconds, while the marker detector takes about 10 milliseconds. Therefore, we proposed a hybrid tracking approach that combines object detection with template matching to enable the tracking of multiple objects with very low processing time. This approach exhibits good tracking performance for objects with drastic appearance changes due to the constantly updated object templates. Due to the low latency of high frame rate (HFR) videos, the motion speed of objects between frames is relatively low. To quickly locate objects near their image blocks, template matching can be utilized. Figure 2 illustrates the hybrid detection method based on template matching and object detection. The time interval between input HFR images is denoted as τ milliseconds. The detector continuously performs object detection with a time interval of δ milliseconds, where δ(δ>τ) represents the processing time of the detector. The detection results $D(I_{t})$ obtained from the detector in the input image $I_{t}$ at time t=k×δ(k=0,1,2,⋯) can be expressed as follows:(1)$$D\left(I_{t}\right)=\left\{d_{t}^{1},d_{t}^{2},\cdots ,d_{t}^{l},\cdots ,d_{t}^{L}\right\}\left(l=1,2,\cdots ,L\right).$$

Each detection result $d_{t}^{l}$ comprises six parameters:(2)$$d_{t}^{l}=\left\{x_{l},y_{l},w_{l},h_{l},p_{l},c_{l}\right\}.$$

$x_{l}$, $y_{l}$, $w_{l}$, and $h_{l}$ represent the starting image coordinates, width, and height of the *l*-th object image block, respectively. $p_{l}$ and $c_{l}$ represent the confidence score and category of the detection result, respectively. As indicated in Figure 2, we obtained object templates $T_{t}$ updated at time intervals of δ.

Simultaneously, we perform template matching using the most recently updated templates to detect objects at time intervals of τ. In high-speed visual systems where the system’s operational speed is a priority, a trade-off between speed and accuracy is often necessary. Therefore, we employ the sum of absolute differences (SAD) as the similarity metric for image-template matching. The detection process for objects between adjacent HFR frames is as follows:(3)$$P_{l}\left(t\right)=P_{l}\left(t-\tau \right)+\underset{\left|x\right|\leq R,\left|y\right|\leq R}{\arg\min}E\left(x,y\right),$$
(4)$$E\left(x,y\right)=\sum_{x^{\prime },y^{\prime }}\left(T_{l}\left(x^{\prime },y^{\prime }\right)-I_{t}\left(x_{t}^{\prime }+x+x^{\prime },y_{t}^{\prime }+y+y^{\prime }\right)\right).$$

$P_{l}\left(t-\tau \right)$ and $P_{l}\left(t\right)$ represent the coordinates of the center of the *l*-th object in the previous and current frames, respectively. $T_{l}$ is the template image of the *l*-th object that is most recently updated. $I_{t}$ represents the region of interest (ROI) being searched in the current image, as highlighted in yellow in Figure 2. $\left(x^{\prime }t,y^{\prime }t\right)$ represents the top-left point coordinate of the ROI region in the current image. *R* is the search range of the template matching. To mitigate the impact of object appearance changes on tracking, we perform template updates by searching in a larger region each time, as depicted in the yellow area in the figure.

In this work, we employ a distance matrix Φ between *I* objects in the previous frame and *J* objects in the current frame as a replacement for the Intersection over Union (IOU) method for object tracking.
(5)$$\Phi =\left[\begin{array}{cccc} \psi (1,1) & \psi (1,2) & \cdots & \psi (1,J)\\ \psi (2,1) & \psi (1,2) & \cdots & \psi (2,J)\\ \cdots & \cdots & \cdots & \cdots \\ \psi (I,1) & S(I,2) & \cdots & \psi (I,J) \end{array}\right].$$ψ(i,j) represents the Euclidean distance between the *i*-th object in the previous frame and the *j*-th object in the current frame, measured in pixels. We employ the Hungarian matching algorithm to obtain tracking results quickly and efficiently.

In high-speed imaging, where object motion is relatively slow and motion between adjacent frames is approximately uniform, we use a Kalman filter for optimal estimation of motion. The Kalman filter can also be used for short-term motion prediction when object detection is temporarily lost.

### 3.3. Correspondence Based on High Synchronous Velocities

#### 3.3.1. Velocity-Based Correspondence

Once the optimal tracking state of the object is obtained, we can obtain highly synchronized spatiotemporal velocities (STVs). As shown in Figure 3, we sampled the velocities of the object at the pixel scale within *N* high-speed frames to extract the motion feature of the object. The STVs *V* of the object were then obtained as follows:(6)$$V=\left\{v_{N-1},\cdots ,v_{n},\cdots ,v_{1},v_{0}\right\},\left(n=0,1,\cdots ,N-1\right),$$
where $v_{n}=\left[dx_{n},dy_{n}\right]$ is the velocity vector at the pixel scale in the *n*-th frame before the current frame.

In this study, we propose the concept of the scale cosine distance. While the calculation of the cosine distance yields the cosine of the angle between both vectors, which is close to 1 when the angle is small, the cosine distance does not consider the length of the vector. This means that two parallel vectors with different lengths would have a cosine distance of 1, even though their similarity is very low. To overcome this limitation, we introduce the scale cosine distance *s* between vectors *A* and *B*, which takes into account the length of the vector, as expressed below:(7)$$s=\frac{A\cdot B}{\max{\left(|A|,|B|\right)}^{2}},$$
where |A| and |B| are the modulo lengths of vectors *A* and *B*, respectively. When the lengths of both vectors are similar and the included angle is small, the scale cosine distance is larger, with a higher similarity close to 1.

Hence, for the *N*-dimensional high-synchronization STVs ${}^{l}V_{i}$ and ${}^{r}V_{j}$ extracted from the left and right HFR stereo cameras, we calculated the scale cosine similarity $S_{v}\left(i,j\right)$ between them as follows:(8)$$S_{v}\left(i,j\right)=\frac{1}{N}\sum_{i=0}^{N-1}\frac{{}^{l}v_{k}\cdot^{r}v_{k}}{\max{\left(|{}^{l}v_{k}{\left|,\right|}^{r}v_{k}|\right)}^{2}}.$$

#### 3.3.2. Direction-Based Correspondence

The correlation of velocity decreases in the presence of a large viewing angle in the HFR stereo camera. The correlation between the direction of velocity change and the change in camera viewing angle is relatively small. We extract the cosine values of the angle changes between velocities to form a short-term angle for measuring the similarity *A* of direction changes.
(9)$$A=\left\{a_{N-2},\cdots ,a_{n},...,a_{1},a_{0}\right\},\left(n=0,1,\cdots ,N-2\right).$$$a_{n}$ is the cosine value between adjacent velocity angles,
(10)$$a_{n}=\frac{v_{n+1}\cdot v_{n}}{|v_{n+1}\left|\cdot \right|v_{n}|},\left(n=0,1,\cdots ,N-2\right).$$

Hence, for the (N−1)-dimensional high-synchronization STVs ${}^{l}A_{i}$ and ${}^{r}A_{j}$ extracted from the left and right HFR stereo cameras, we calculated the direction similarity $S_{a}\left(i,j\right)$ between them as follows:(11)$$S_{a}\left(i,j\right)=1-\frac{1}{2(N-1)}\sum_{i=0}^{N-2}|{}^{l}a_{k}-{}^{r}a_{k}|.$$

#### 3.3.3. Mixed Correspondence

The similarity measure of object motion is contributed by both the similarity of velocities and the similarity of velocity change directions. We define the mixed similarity S(i,j) between the short-term velocities of the *i*-th target in the left camera and the *j*-th target in the right camera as follows:(12)$$S\left(i,j\right)=\omega_{v}S_{v}\left(i,j\right)+\omega_{a}S_{a}\left(i,j\right),$$
(13)$$\;s.t.\;\omega_{v}+\omega_{a}=1.$$
where $\omega_{v}$ and $\omega_{a}$ are scale factors that reflect the contribution of velocity and direction to the similarity metric in different camera perspectives. Generally, when the HFR stereo camera has a large field of view, the direction similarity $S_{a}\left(i,j\right)$ should contribute a larger proportion. Finally, based on the mixed similarity of short-term velocities, a bipartite graph *S* can be reconstructed for *I* targets in the left camera and *J* targets in the right camera,
(14)$$S=\left[\begin{array}{cccc} S(1,1) & S(1,2) & \cdots & S(1,J)\\ S(2,1) & S(1,2) & \cdots & S(2,J)\\ \cdots & \cdots & \cdots & \cdots \\ S(I,1) & S(I,2) & \cdots & S(I,J) \end{array}\right].$$

Using the Hungarian matching algorithm, we can easily obtain the correspondence relationship based on motion information.

## 4. Experiment

The proposed stereo correspondence algorithm was implemented offline using an HFR stereo camera system that operated at a speed of 200 fps. The system was composed of two high-speed USB 3.0 camera heads from Imaging Source Corp. (DFK 37BUX273, Germany) and a personal computer. The cameras were compact, measuring 36×36×25 mm in size, weighing 70 g, and had no mounted lens. They were capable of capturing and transferring 10-bit color images of 1440×1080 pixels to RAM at a rate of 238 fps via a USB 3.0 interface. We used a PC with the following hardware specifications to record the HFR stereoscopic video: Intel Core i9-9900K @ 3.2 GHz CPU, 64 GB RAM, and an NVIDIA GeForce RTX 2080 Ti GPU.

To evaluate the performance of our stereo correspondence algorithm, we analyzed HFR stereo offline videos that were captured at a rate of 200 fps (τ=5 ms) with a 2-ms exposure time. In this study, we chose the hand as the detection target because it had a high similarity in texture and color across different people, and moved at a high speed relative to other body parts, making it difficult to use appearance-based methods for correspondence. We conducted three experiments to evaluate our algorithm: stereo correspondence evaluation, correspondence of fast-moving hands, and correspondence in a meeting room scene. For the hand detection task, we used Google’s MediaPipe toolkit, which provided accurate and rapid hand detection.

### 4.1. Stereo Correspondence Evaluation

We conducted an evaluation of the correspondence performance of our HFR stereo correspondence algorithm when implemented offline in our system. Figure 4 illustrates the experimental setup for the stereo correspondence evaluation, where two metronomes were fixed 800 mm away from the HFR stereo camera. The small metronomes operated at frequencies of 3.0 and 2.6 Hz, respectively. OpenCV-generated markers were attached to different positions on the pointers of both metronomes. As a result, markers on a similar pointer exhibited similar movements when shaking, but with different magnitudes of movement. During the operation of the metronomes, we captured a 200-fps HFR stereoscopic video using 12-mm lens fixed cameras. The positions of the individual markers were easily detected using OpenCV.

In Figure 5, we show the input stereo images of size 1440×1080 pixels, with the correspondence results at intervals of 0.06 s for *t* = 7.00∼7.25 s. After applying the stereo correspondence algorithm, the same marker in the HFR stereoscopic video was marked with numerical symbols of a similar color. The xy coordinate values of the image centroids of the markers in the left HFR stereoscopic video are presented in Figure 6. From the image, markers 0 and 1 exhibited similar movement with different magnitudes than markers 2 and 3. The mixed similarities of the moving markers’ STVs over time are shown in Figure 7. Figure 7a–d depict the mixed similarities between markers 0, 1, 2, and 3 in the left HFR stereo image and those in the right HFR stereo images, respectively. The graph indicates that similar markers in the HFR stereo images have a high degree of similarity, which is almost greater than 0.8. Markers 0 and 1 on a similar pointer have a similar angular velocity, but different linear velocities. However, our scale cosine distance includes a scale factor that can easily distinguish between markers 0 and 1. The same applies to markers 2 and 3. We also considered the effect of the duration of STVs on multi-object stereo correspondence. Figure 8 presents the results of stereo correspondence using a 30 fps stereo camera in the same scene. It is evident that marker 0 and marker 3 do not match in the correspondence. Figure 9 shows the short-term velocity features of marker 0 within a 0.3-second interval in the stereo camera at *t* = 7.710 s. It is evident that traditional low-speed cameras have synchronization issues when tracking fast-moving objects. Velocity information is delayed by approximately 30 milliseconds, which significantly affects the correspondence results, especially when the object changes direction frequently. Figure 10 shows the short-term velocity features of marker 0 within a 0.3-second interval in the HFR stereo camera. In contrast, the HFR camera not only provides more motion information in a short time but also has much higher synchronization.

### 4.2. Stereo Correspondence of Hands with Complex Movements

We present the stereo correspondence results of hand movements during complex actions such as overlap and reappearance. The experimental setup is illustrated in Figure 11. Two individuals waved their hands approximately 8 m away from the HFR stereo cameras. Similar to the previous experiment, we captured a 200-fps HFR stereoscopic video using 12-mm fixed lens cameras. The hand movements in the video included mutual occlusion, static states, disappearance, and reappearance. There were four hands in the HFR stereoscopic video, represented by hand 2, hand 0, hand 1, and hand 3 from left to right. In the offline detection process, we utilized MediaPipe to detect the hands.

In Figure 12, we depict the input HFR stereo images with a resolution of 1440×1080 pixels and the correspondence results at intervals of 0.05 s for *t* = 12.84∼13.09 s. In the HFR stereoscopic video, similar hands are marked with similar colors from left to right. As shown in the graph, there is an overlap between hands 2 and 0, which belong to the person on the left. Hands 0 and 1, belonging to different people, also overlap. Our method correctly predicts the position of the hands and completes the hand correspondence even in the case of missing objects. The xy coordinate values of the image centroids of the hands in the left HFR stereoscopic video are shown in Figure 13. By analyzing the trajectories of the four hands, we can decompose the entire motion process into multiple actions. From 1.8 to 9.0 s, the hands belonging to the same person crossed each other and moved. From 23.0 to 28.0 s, hands 2 and 3 disappeared and reappeared. For the rest of the time, the four hands were stationary. The mixed similarities of different hands’ STVs over time are shown in Figure 14. Figure 14a–d show the mixed similarities of STVs between hands 0, 1, 2, and 3 in the left HFR stereo image and those in the right HFR stereo images, respectively. Similar to the metronome correspondence, the motion features of a similar hand in the HFR stereoscopic video have a higher similarity. Since our features are motion-based, it can be seen from the figure that missing motion features introduced more uncertainty when the hand was stationary. Furthermore, we added appearance-based correspondence methods and calculated the accuracy of each method for hand correspondence every 0.25 s, as shown in Figure 15. The deep learning methods, ResNet and DeepCompare, achieved significantly better results throughout the process and were clearly superior to traditional methods. Our method maintained an accuracy of almost 100% during hand movement. The accuracy rate was lower than that of the appearance-based methods only when the hand was stationary. In calibrated stereo cameras, when similar objects are found in the stereo camera, their spatial positions can be calculated. In Figure 16, we plotted the 3D trajectories of hands 0, 1, 2, and 3 over 7 to 8 s. From the image, we can see that the four hands moved up and down at a distance of approximately 8 m from the camera. The acquisition of spatial information helped us to better analyze the movement of objects.

### 4.3. Stereo Correspondence in the Meeting Room

Finally, we present the experimental results for stereo correspondences when the stereo cameras operate at 200 fps in a meeting room. To obtain a larger field of view, the stereo cameras are equipped with 6-mm lenses. The experimental setup is illustrated in Figure 17. In the meeting room, several students were more than 2 m away from the stereo cameras. Due to factors such as privacy and occlusion, it was difficult to detect and identify different students by their faces. Obtaining the spatial position using stereo correspondence of the hands is a feasible solution to identify different students.

In Figure 18, we depict the input HFR stereo images of 1440×1080 pixels with correspondence results at intervals of 0.1 s for *t* = 7.180∼7.680 s. We numbered the students from 1 to 5, from the nearest to the farthest. Similar hands in the stereo HFR video were marked with similar colors, as in the previous experiment. When the students raised their hands, we performed stereo correspondence using hand movements. Furthermore, we calculated the 3D positions of the different hands. In this experiment, we knew the seating distribution of each student in advance, and we could identify who raised their hand through the position of the hand. In Figure 18, we marked the hand-raising action of classmates in the upper right corner. When the hands were raised, circles belonging to different students were filled with different colors; otherwise, they were filled with black. The xy coordinate values of the hand images in the left HFR stereoscopic video are shown in Figure 19 at *t* = 0–16 s. Simultaneously, Figure 20 shows the time variation of the mixed similarity between similar hands at *t* = 0–16 s. From the graph, the hands of students 1 to 5 appeared individually in the HFR stereoscopic video. The students’ hands moved at 0–7 and 10.5–16 s. During motion, the same hand in the HFR stereoscopic video had a high mixed similarity of approximately 0.8. We stopped the hand from moving at 6.8–10.2 s. As seen in Figure 20, the mixed similarities of the same hand dropped rapidly, greatly reducing the accuracy of the correspondence. Figure 21 shows the Gantt chart of the detected students’ hands raised over time. When the hand stopped moving, we could not accurately complete the correspondence. Our algorithm is currently limited regarding stereo correspondence in the static state. These results show that our method can accurately match objects in a stereoscopic video in moving scenes and use spatial information to complete certain applications.

## 5. Conclusions

In this study, we addressed the problem of stereo correspondence of objects with similar appearances. Traditional appearance-based algorithms do not provide effective performance, so we proposed a method that uses highly synchronized motion information to overcome this limitation. Our approach involves using high-synchronous short-term velocities acquired by high-speed vision systems as features for stereo correspondence of moving objects. We demonstrated the effectiveness of our method through experiments on (1) the correspondence of markers for regular motion on a metronome and (2) motion tracking and correspondence of multiple hands in indoor scenes. These experiments confirmed the potential of high-speed vision technology to improve the stereo correspondence of objects with similar appearances. However, our current method cannot provide accurate results when objects are static.

In the future, we aim to address the following issues to further improve our method: (1) We will add appearance-related factors to our algorithm to achieve higher accuracies in stereo correspondences when objects are static. (2) Currently, our algorithm is run offline, but we plan to implement it using real-time HFR stereo camera systems in the future.

## Figures and Tables

**Figure 1 sensors-23-04285-f001:**
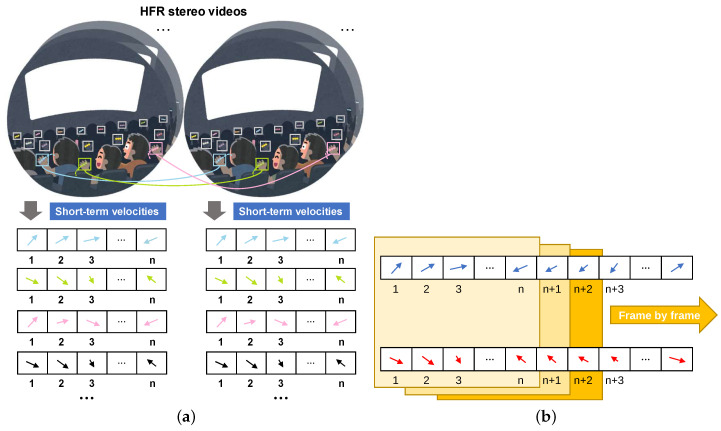
Concept of stereo correspondence based on high synchronous short-term velocities. (**a**) Motion features composed of short-term velocities. (**b**) Correspondence based on short-term velocities.

**Figure 2 sensors-23-04285-f002:**
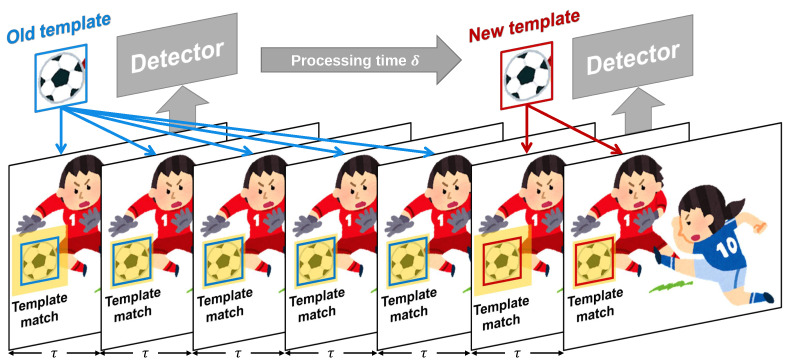
Hybrid detection method based on template matching and object detector.

**Figure 3 sensors-23-04285-f003:**
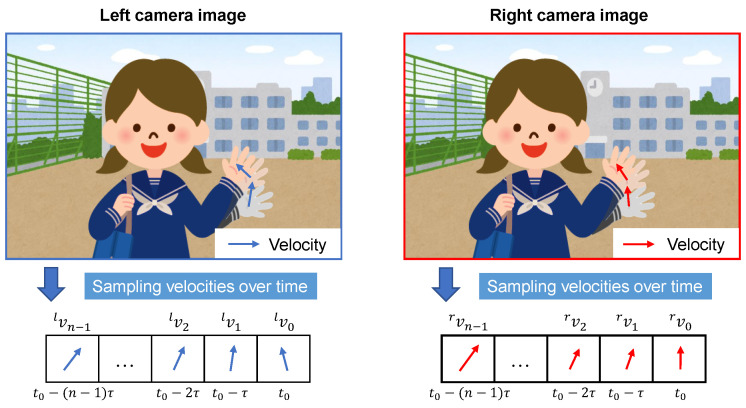
Sampling velocities over time in HFR stereoscopic video.

**Figure 4 sensors-23-04285-f004:**
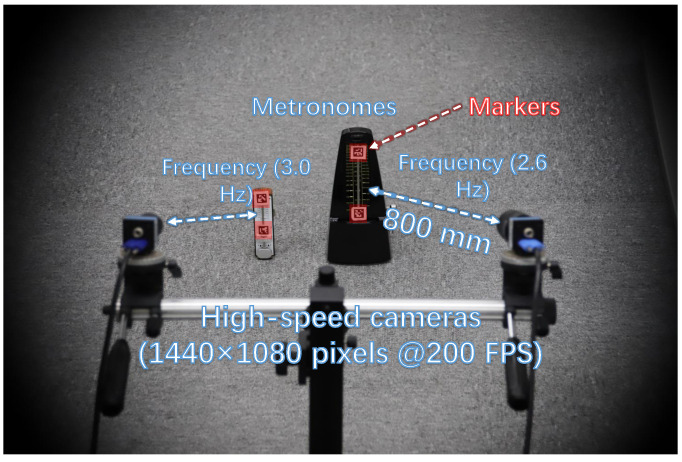
Experiment setup for similar motion correspondence.

**Figure 5 sensors-23-04285-f005:**
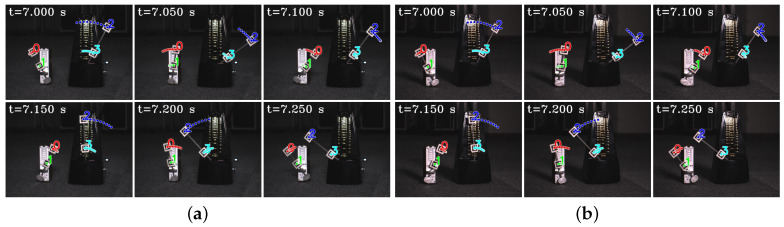
Input images and correspondence result in evaluation. (**a**) Left HFR stereoscopic video. (**b**) Right HFR stereoscopic video.

**Figure 6 sensors-23-04285-f006:**
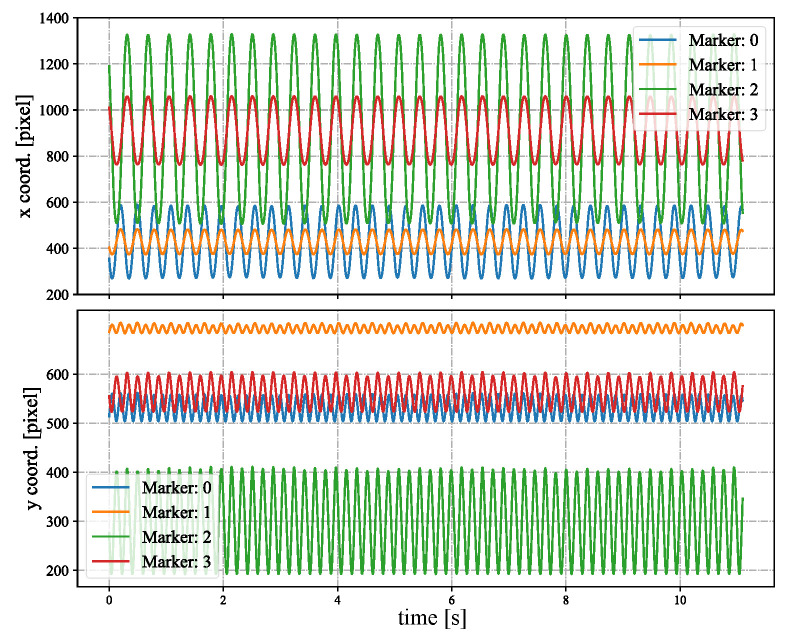
Image centroids of the markers in the stereo correspondence evaluation.

**Figure 7 sensors-23-04285-f007:**
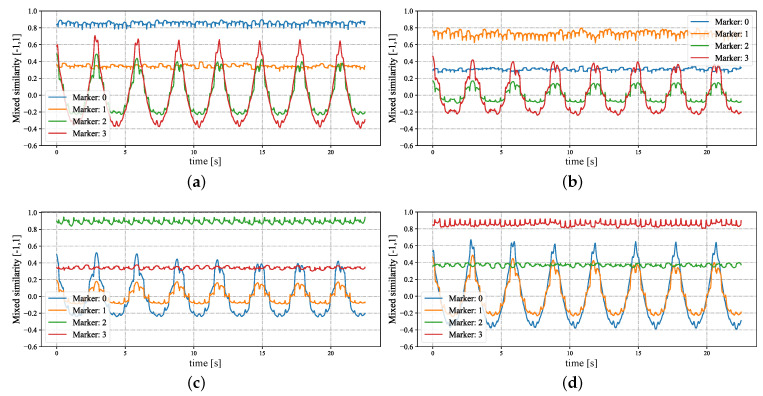
Mixed similarities of different markers in the HFR stereoscopic video when the STVs length is 64. (**a**) Marker 0 in the left video. (**b**) Marker 1 in the left video. (**c**) Marker 2 in the left video. (**d**) Marker 3 in the left video.

**Figure 8 sensors-23-04285-f008:**
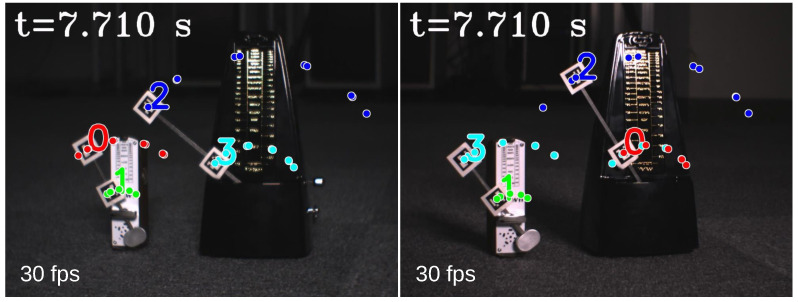
Correspondence results using a stereo camera at 30 fps (t = 7.710 s).

**Figure 9 sensors-23-04285-f009:**
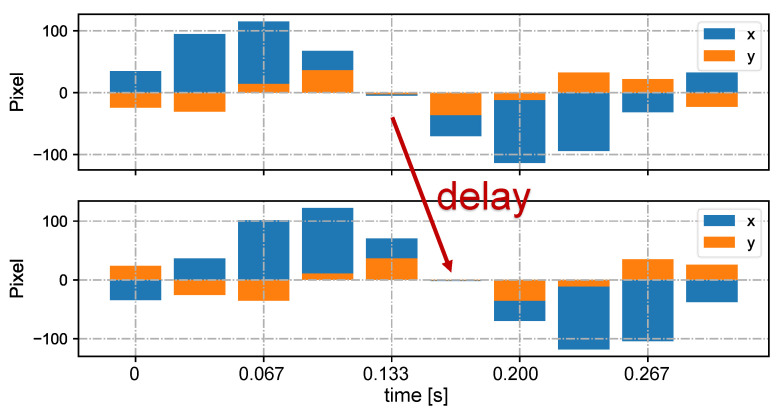
Short-term velocities of marker 0 in the stereo video in 0.3 s at 30 fps (t = 7.710 s).

**Figure 10 sensors-23-04285-f010:**
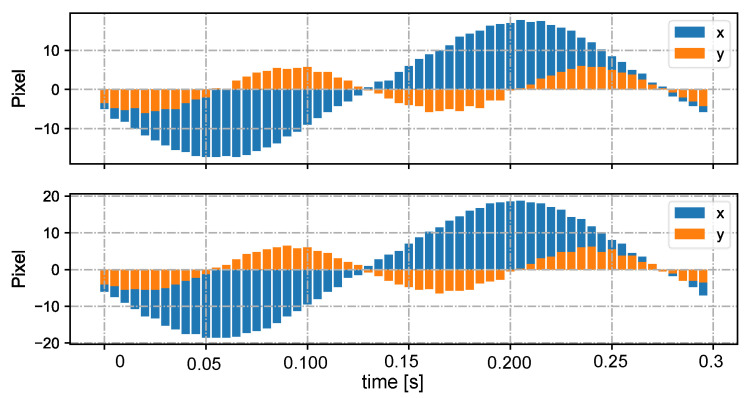
Short-term velocities of marker 0 in the stereo video in 0.3 s at 200 fps.

**Figure 11 sensors-23-04285-f011:**
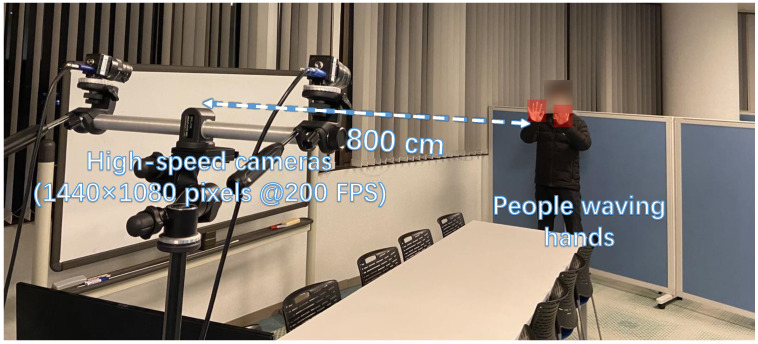
Experiment setup for hand stereo correspondence.

**Figure 12 sensors-23-04285-f012:**
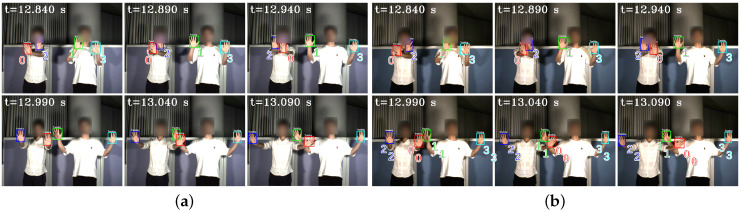
Input images and hand correspondence result. (**a**) Left HFR stereo images. (**b**) Right HFR stereo images.

**Figure 13 sensors-23-04285-f013:**
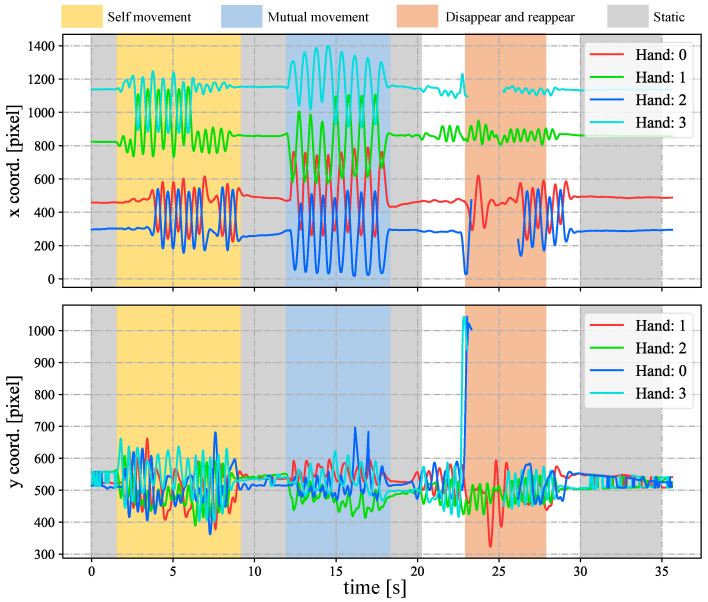
Image centroids of the hands in the left HFR stereoscopic video.

**Figure 14 sensors-23-04285-f014:**
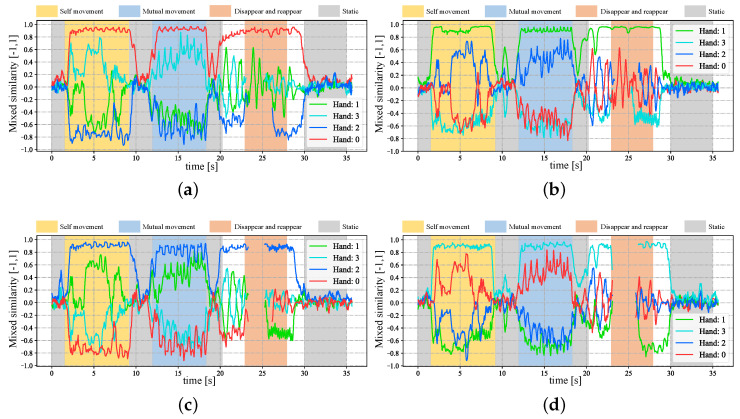
Mixed similarities between different hands in the HFR stereoscopic video when the STVs length is 64. (**a**) Hand 0 in the left video. (**b**) Hand 1 in the left video. (**c**) Hand 2 in the left video. (**d**) Hand 3 in the left video.

**Figure 15 sensors-23-04285-f015:**
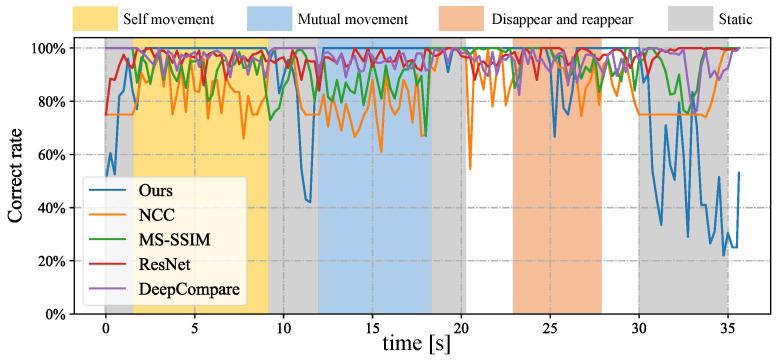
Correct rate of different stereo correspondence methods updated every 0.25 s.

**Figure 16 sensors-23-04285-f016:**
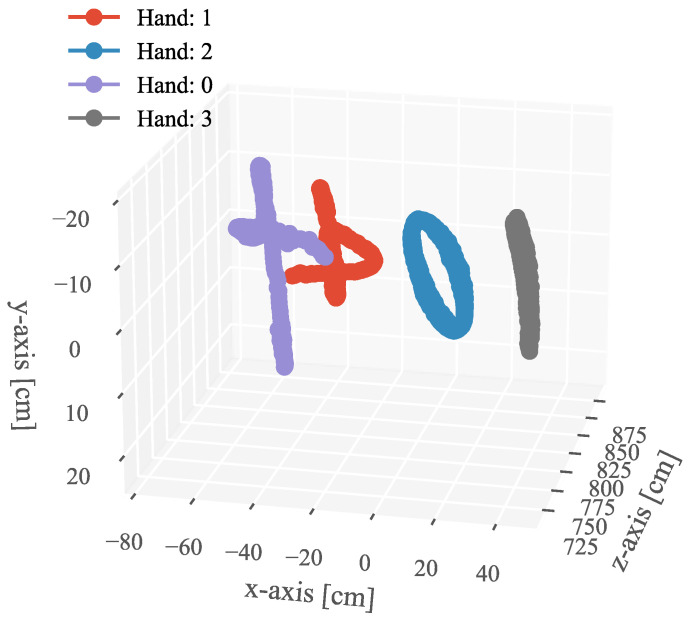
3D trajectory of each hand with 7∼8 s.

**Figure 17 sensors-23-04285-f017:**
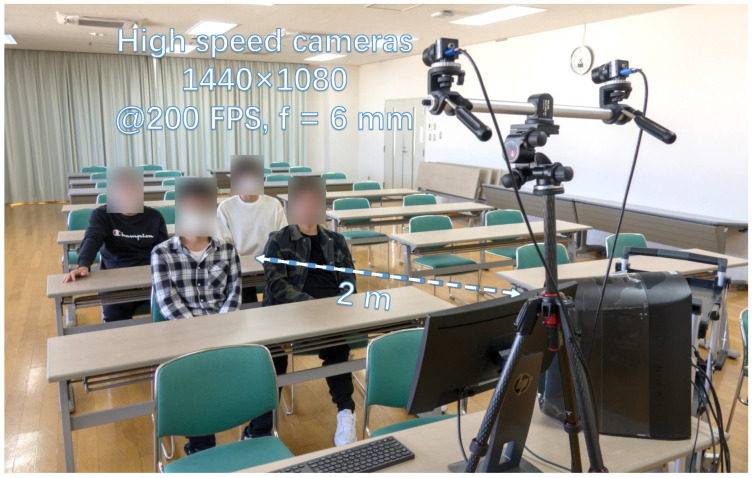
Experimental environment for stereo correspondence in the meeting room.

**Figure 18 sensors-23-04285-f018:**
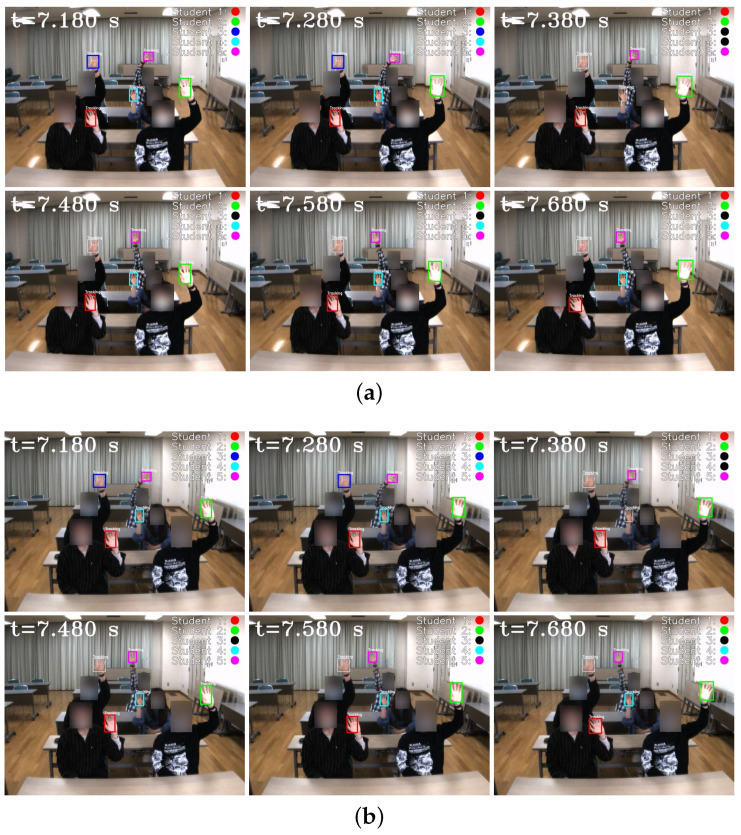
Input images and stereo correspondence result. (**a**) Left HFR stereo images. (**b**) Right HFR stereo images.

**Figure 19 sensors-23-04285-f019:**
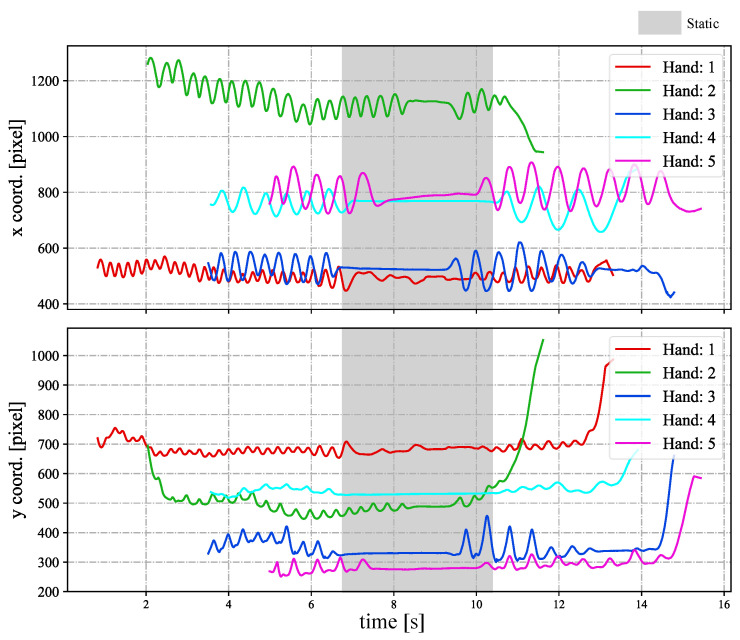
Image centroids of hands in the left HFR stereoscopic video.

**Figure 20 sensors-23-04285-f020:**
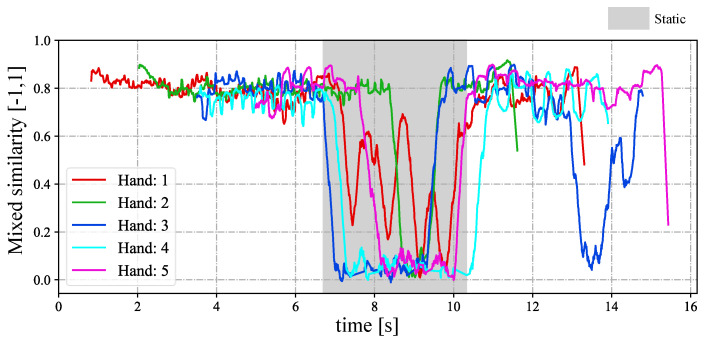
Mixed similarities between a similar hand in the HFR stereoscopic video when the STVs length is 64.

**Figure 21 sensors-23-04285-f021:**
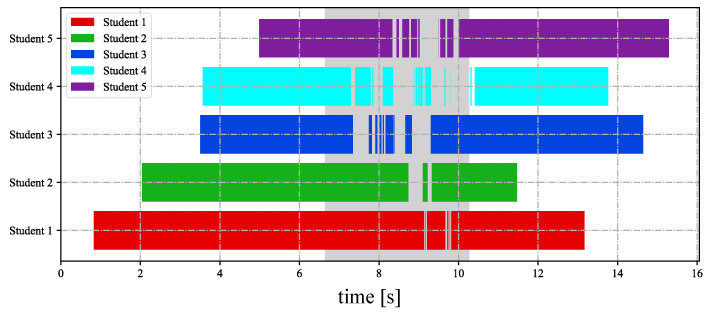
Statistical analysis of raised hands.

## Data Availability

Data sharing not applicable to this article as no datasets were generated or analysed during the current study.

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
