# Peer review of "HFR-Video-Based Stereo Correspondence Using High Synchronous Short-Term Velocities"

_sensors, 2023, doi:10.3390/s23094285_

Round 1
Reviewer 1 Report
This paper describes the correspondence matching (useful for stereo depth estimation), done purely on short term motion features, acquired using high speed cameras.
I have to say I like the paper, because the idea sounds like something that needed to be done, and that's why i accepted the review task. But nevertheless, I have some criticism, the paper could be improved significantly.
1. References. There are many old references. Was one of the authors fresh phd student or undergrad? You really don't need to cite Kalman filter. Please clean up your references and, if possible, add some recent ones that are closely related to your work. Additionally, check arXiv references, if there are newer publications on the same topic from same authors that have been peer reviewed, and if so, update the reference!
2. Tracking. Section 3.2 and especially 3.2.1 are confusing. My opinion was that the description is too low on detail, before I read later that you used some other algorithm for tracking, that you did not develop it yourself. Please rewrite 3.2 in the manner that makes this more obvious. Limit yourself on your contribution only. How is Kalman that you describe in 3.2.2 different from the usual approach, if at all? If it is different, emphasize the differences.
Otherwise, paper reads smoothly.
Author Response
Response to Reviewer 1 Comments
Point 1: References. There are many old references. Was one of the authors fresh phd student or undergrad? You really don't need to cite Kalman filter. Please clean up your references and, if possible, add some recent ones that are closely related to your work. Additionally, check arXiv references, if there are newer publications on the same topic from same authors that have been peer reviewed, and if so, update the reference!
Response 1: Thank you very much for your suggestion. This is my first time writing a journal paper, so there are many deficiencies. When citing papers, I tend to cite more classic papers at first. Thank you for your corrections, I have revised the references and cited many papers with new application directions.
Point 2: Tracking. Section 3.2 and especially 3.2.1 are confusing. My opinion was that the description is too low on detail, before I read later that you used some other algorithm for tracking, that you did not develop it yourself. Please rewrite 3.2 in the manner that makes this more obvious. Limit yourself on your contribution only. How is Kalman that you describe in 3.2.2 different from the usual approach, if at all? If it is different, emphasize the differences.
Response 2: Thank you very much for your suggestion. The tracking section is really poorly written. I thought at first that tracking was not the point of this tracking, and I have reworked the paper based on your suggestion. In this paper, in order to track objects in offline high-speed video, we use a tracking method mixed with template matching. The detection algorithm used later in this article includes the QR code detection that comes with OpenCV and the MediaPipe model of Google. These algorithms are difficult to achieve real-time detection within the high-speed frame update time. We use the object template provided by the detector for regional template matching to achieve fast detection. Then use the shortest distance from the center of the image instead of the IOU to track the target. As for the Kalman filter, I used it to reduce the jitter error caused by the detection, and did not improve it. So the use of the Kalman filter is just mentioned in the revised manuscript.
Figure 2. Hybrid detection method based on template matching and object detector.

Reviewer 2 Report
The presented paper regards with actual problem of recognizing the multiple similar objects in the scene correctly.
The Introduction and Related work chapters are very well structured and they give a good overview of discussed problematic to the reader.
The symbol L in equation (1) in not explained – probably the number of objects detected in the scene? Further, if the superscript of detector d in (1) starts form 1, the L starting from 0 makes no sense.
Line 158: the word “confidence” should be probably used instead of the word “conference”. Perhaps the spellchecker action.
Lines 172, 173: The left superscript m meaning is not explained.
Line 197, 217: The first letter in the sentence should be a capital one.
What types of markers were used in case of the experiment with metronomes?
Figure 9 is missing.
The proposed algorithm is compared with another types of algorithms only in the case of the hands with complex movements. It would be beneficial if a similar comparison was carried out also for experiment with metronomes and in the meeting room.
Despite the minor errors mentioned above I find the proposed article as beneficial.
Author Response
Response to Reviewer 2 Comments
Point 1: The symbol L in equation (1) in not explained – probably the number of objects detected in the scene? Further, if the superscript of detector d in (1) starts form 1, the L starting from 0 makes no sense.
Response 1: Thank you very much for your suggestion. Here, the number of different detection targets is subconsciously described in a programming language, which has been modified as follows,
The detector will detect objects at a time, and d represents the -th object.
Point 2: Line 158: the word “confidence” should be probably used instead of the word “conference”. Perhaps the spellchecker action.
Response 2: Thank you very much for your suggestion. This should be a typo, it is indeed the confidence level. I have fixed this problem.
Point 3: Lines 172, 173: The left superscript m meaning is not explained.
Response 3: Thank you very much for your suggestion. There should be a typo here. I originally wanted to use the on the upper left to represent the label of the stereo camera. But in the subsequent revision of the paper, I have deleted the process of updating the Kalman filter.
Point 4: Line 197, 217: The first letter in the sentence should be a capital one.
Response 4: Thank you very much for your suggestion. This was a compilation error and became a paragraph in the template provided by MDPI, and I have revised the paper.
Point 5: What types of markers were used in case of the experiment with metronomes?
Response 5: Thank you very much for your suggestion. I used the Aruco library that comes with OpenCV to draw the marker. Aruco marker is a kind of marker similar to a two-dimensional code. By inputting the size of the marker, it can detect the marker in the image and calculate the 3D position of the marker in the camera coordinate system. In this work, I pasted the ID as 0, 1, 2, 3 marks.
Figure 1. Aruco marker
Point 6: Figure 9 is missing.
Response 6: Thank you very much for your suggestion. I didn't understand what you mean. Figure 9 in the paper should be at the bottom of the paper. This image mainly shows a stereoscopic video with a traditional frame rate, and there are certain deficiencies in synchronization. This affects the matching between motion information.
Point 7: The proposed algorithm is compared with another types of algorithms only in the case of the hands with complex movements. It would be beneficial if a similar comparison was carried out also for experiment with metronomes and in the meeting room.
Response 7: Thank you very much for your suggestion. The corresponding experiment of the first metronome in the paper is mainly to demonstrate the performance of our high-speed system for high-speed complex moving objects. As mentioned by Point 5 above, I posted 4 markers with different IDs, and there are big differences in appearance. And the view angle between the cameras is small, and it can be easily distinguished by using the similarity algorithm based on appearance, so the comparison data is not placed in the paper.

Round 2
Reviewer 1 Report
Paper is fine, I would just suggest spellchecking and minor English corrections (plural/singular words), there are some superscripts "Li" that are probably some leftover from markup to show the differences... they have to be removed.
Author Response
Response to Reviewer 2 Comments
Point 1: Paper is fine, I would just suggest spellchecking and minor English corrections (plural/singular words), there are some superscripts "Li" that are probably some leftover from markup to show the differences... they have to be removed.
Response 1: Thank you very much for your suggestion. The superscript of "Li" is to mark the part I modified, to help you better find the location of my modification. In the new version of the paper, I have removed the markup. And based on your suggestion, I re-examined the grammatical part of the paper, including singular and plural. Thank you very much for your review.
